# Association between the Respiratory Microbiome and Plasma Microbial Extracellular Vesicles in Intubated Patients

**DOI:** 10.3390/microorganisms11092128

**Published:** 2023-08-22

**Authors:** Jinkyeong Park, Seong Ji Woo, Yoonki Hong, Jae Jun Lee, Ji Young Hong

**Affiliations:** 1Department of Pulmonary, Allergy and Critical Care Medicine, Kyung Hee University Hospital at Gangdong, School of Medicine, Kyung Hee University, Seoul 05278, Republic of Korea; pjk3318@gmail.com; 2Institute of New Frontier Research Team, Hallym University College of Medicine, Chuncheon 24253, Republic of Korea; seong-jikr@nate.com (S.J.W.); iloveu59@hallym.or.kr (J.J.L.); 3Department of Internal Medicine, Kangwon National University Hospital, School of Medicine, Kangwon National University, Chuncheon 24289, Republic of Korea; h-doctor@hanmail.net; 4Division of Pulmonary, Allergy and Critical Care Medicine, Department of Internal Medicine, Chuncheon Sacred Heart Hospital, Hallym University Medical Center, Chuncheon 24253, Republic of Korea

**Keywords:** extracellular vesicle, microbiome, communication

## Abstract

Extracellular vesicles (EVs) regulate various cellular and immunological functions in human diseases. There is growing interest in the clinical role of microbial EVs in pneumonia. However, there is a lack of research on the correlation between lung microbiome with microbial EVs and the microbiome of other body sites in pneumonia. We investigated the co-occurrence of lung microbiome and plasma microbe-derived EVs (mEVs) in 111 samples obtained from 60 mechanically ventilated patients (41 pneumonia and 19 non-pneumonia cases). The microbial correlation between the two samples was compared between the pneumonia and non-pneumonia cases. Bacterial composition of the plasma mEVs was distinct from that of the lung microbiome. There was a significantly higher correlation between lung microbiome and plasma mEVs in non-pneumonia individuals compared to pneumonia patients. In particular, Acinetobacter and Lactobacillus genera had high correlation coefficients in non-pneumonia patients. This indicates a beneficial effect of mEVs in modulating host lung immune response through EV component transfer.

## 1. Introduction

Extracellular vesicles (EVs) are small particles released from eukaryotic host cells or pathogens [1]. They have emerged as novel mechanisms for cell-to-cell communication [2], including the transfer of EVs derived from bacteria, which can modulate a host’s innate immunity [3]. EVs are known to be enriched with proteins, lipids DNA and miRNAs [4]. These EV-containing miRNAs are transferred to specific cells and regulate pathogenic processes such as angiogenesis, coagulation and inflammation [5]. EVs released by respiratory pathogens facilitate pro-inflammatory responses of pulmonary epithelium and various immune cells [6]. In parallel, they can induce robust antibody response and reduce bacterial replications, making them promising vaccine candidates [6].

Investigating the microbiome EVs across different body sites can provide comprehensive insights into disease mechanisms and the influence of environmental factors. Recently, increased transfer of gut microbe-derived EVs (mEVs) to the blood has been implicated in the pathogenesis of type 2 diabetes mellitus (T2DM) [7]. However, little is known about the association between plasma mEVs and the respiratory microbiome, particularly in respiratory diseases such as pneumonia. Multiple studies have shown that gut microbiota play an important role in the immune response of the lungs. Additionally, it has been observed that gut mEVs enter the bloodstream, triggering various immune and metabolic reactions [8,9,10,11]. However, it remains unclear whether the respiratory microbiome crosses the alveolar capillary barrier to enter the blood, or if plasma EVs have an effect on the lung microbiome. Further studies are required to clarify the relationship between the respiratory microbiome and plasma mEVs. Therefore, this study analyzed plasma mEVs and respiratory microbiomes in intubated patients and explored the microbial correlation between the two samples. Identifying the association and differences in genomic profiling between respiratory microbiome and plasma mEVs may provide insights into the pathogenesis of pneumonia and the clinical role of plasma mEVs.

## 2. Materials and Methods

### 2.1. Study Subjects

This prospective study was conducted at the ICU of Chuncheon Sacred Heart Hospital, South Korea. In total, 120 samples were obtained from 60 subjects who were placed on mechanical ventilation at the time of ICU admission between 18 July 2017, and 15 August 2018 (Figure 1). The exclusion criteria were age < 18 years, initiation of mechanical ventilation ≥ 48 h after ICU admission or mechanical ventilation for <7 days, presence of a neuromuscular disease such as amyotrophic lateral sclerosis and (4) inability to provide informed consent. Endotracheal aspirate (ETA) and plasma samples were collected after > 8 h of fasting. The reasons of mechanical ventilation included the cardiac arrest (n = 4), neurosurgical condition (n = 19), postoperative status (n = 1) and respiratory distress (n = 36). The diagnosis of pneumonia was comprehensively made by physical examination, history taking and radiographic criteria such as chest radiography and chest computed tomography [12]. A diagnosis of pneumonia was made in 41 subjects, followed by treatment based on international guidelines [12,13,14]. Clinical information, including demographic characteristics, presence of acute respiratory distress syndrome (ARDS), severity of illness (Acute Physiology and Chronic Health Evaluation [APACHE] II and Sequential Organ Failure Assessment [SOFA] scores) and clinical outcomes (28-day all-cause mortality, in-hospital mortality and duration of mechanical ventilation), was collected. The study was approved by the Institutional Review Board of Chuncheon Sacred Heart Hospital (approval No. 2017-47). Written informed consent was obtained from the participants, and the study was conducted in accordance with the approved guidelines.

### 2.2. EV Isolation and DNA Extraction

ETA samples and plasma samples were collected within 7 days of initiating mechanical ventilation. Plasma samples were obtained using BD Vacutainers (BD, Franklin Lakes, NJ, USA) and then centrifuged at 2000× *g* for 15 min at 4 °C. To isolate the EVs from plasma samples, a microcentrifuge (Labogene 1730R; BMS, Seoul, Republic of Korea) was used to perform differential centrifugation at 10,000× *g* for 10 min at 4 °C to separate the pellets and supernatant. The supernatant was subsequently filtered through a 0.22 µm filter to eliminate bacteria and foreign particles. Then, the EVs were boiled for 40 min at 100 °C and centrifuged for 30 min at 13,000 rpm and 4 °C. EV DNA extraction was carried out using the DNeasy PowerSoil Kit (Qiagen, Hilden, Germany), and quantification was performed using the QIAxpert system (Qiagen) based on previously described methods. For ETA samples, total DNA extraction was performed using a commercial microbial DNA isolation kit (Qiagen, Hilden, Germany).

### 2.3. Paired-End Reads Sequencing and Data Processing

The extracted DNA underwent amplification using primers designed to target the V3 to V4 regions of the prokaryotic 16S rRNA gene. V3–V4 is a widely used primer set, targeting different hypervariable regions of the 16S rRNA gene. The primers were: 16S_V3_F (5′-TCGTCGGCAGCGTCAGATGTGTATAAGAGACAGCCTACGGGNGGCWGCAG-3′) and 16S_V4_R (5′-GTCTCGTGGGCTCGGAGATGTGTATAAGAGACAGGACTACHVG GGTATCTAATCC-3′). The amplicon library was quantified and subjected to sequencing using the MiSeq platform (Illumina, San Diego, CA, USA), following the manufacturer’s instructions. The paired-end reads were merged using the FLASH software (http://www.cbcb.umd.edu/software/flash), and reads shorter than 350 bp or longer than 550 bp were excluded from further analysis. Chimeric sequences and singletons were removed using VSEARCH and the SILVA gold database. Sequencing reads with >97% similarity were clustered into operational taxonomic units using the SILVA128 database. Samples with low read counts (<150) or failed PCR were excluded for quality control.

### 2.4. Statistical Analysis

Of 120 samples collected from 60 subjects, 9 samples were excluded due to low read counts and 111 samples were finally used in the analysis. Biodiversity and community similarity analyses were conducted using R software (Version 4.0.4). Alpha and beta diversities were calculated using QIIME. Beta diversity was assessed using the Bray–Curtis dissimilarity index, while alpha diversity was measured using the Chao1 index. Taxa that accounted for >0.5% of the relative abundance were considered predominant. The correlation between the lung microbiome and plasma mEVs was evaluated using Spearman correlations. A permutation-based *p*-value was calculated from 100,000 permutation replicates, as it is robust for non-normality and small sample sizes. Genera with a correlation coefficient exceeding 0.5 and a false discovery rate less than 0.05 were considered statistically significant.

## 3. Results

The baseline characteristics of the subjects are presented in Table 1. There were 43 men (71.7%) and 41 people (68.3%) over the age of 65. There were 41 pneumonia cases and 19 non-pneumonia cases. There were no significant differences between the pneumonia and non-pneumonia groups in terms of age, sex, Charlson Comorbidity Index, severity of illness (SOFA and APACHE II scores) and clinical outcomes, including mortality and duration of mechanical ventilation. Inflammatory markers such as CRP and Procalcitonin were significantly higher in the pneumonia group than the non-pneumonia group. As expected, the non-pneumonia group exhibited higher PaO_2_/FiO_2_ ratios. Regarding the causes of intubation, neurosurgical conditions and cardiac arrest were more frequently identified in the non-pneumonia group.

### 3.1. Microbial Diversity between Respiratory Microbiome and Plasma Microbial EVs

Alpha diversity was significantly higher in the ETA samples than in plasma mEVs (Figure 2). The beta-diversity analysis using principal coordinate analysis (pCoA) revealed significant differences between the plasma mEVs and ETA microbiomes at both the phylum and genus levels. At the phylum level, 22 strains were found to be common between the ETA microbiome and plasma mEVs. Proteobacteria, Firmicutes, Actinobacteria, Bacteroidetes and Fusobacteria were the predominant taxa, accounting for >0.5% of the relative abundance.

At the genus level, totals of 898 and 351 genera were identified in the ETA microbiome and plasma mEVs, respectively. Of these, 279 genera were detected in both sampling sites. The microbial taxonomic profiles exhibited significant differences between the ETA microbiome and plasma mEVs (Figure 3). The four most abundant phyla in both the ETA microbiome and plasma mEVs were Proteobacteria, Firmicutes, Actinobacteria and Bacteroidetes. However, Streptococcus and Enterobacter were significantly more abundant in the ETA microbiome than in plasma mEVs, while Staphylococcus, Escherichia-Shigella and Klebsiella were significantly more abundant in plasma mEVs. No genus showed a significantly high correlation between the two sites.

### 3.2. The 16S rRNA Microbiome

In both pneumonia and non-pneumonia cases, the composition of plasma mEVs was distinct from that of the ETA microbiome, which is consistent with the aforementioned findings in the total 111 samples. The beta-diversity analysis using PCoA revealed significant differences between the plasma mEVs and ETA microbiome in both the pneumonia and non-pneumonia cases (Figure 4, *p* < 0.001). Taxa that differed in the lung microbiome and plasma mEVs between the pneumonia and non-pneumonia cases were identified. In particular, Corynebacterium and Stenotrophomonas were predominant in the pneumonia group while *Prevotella* and *Alloprevotella* were predominant in the non-pneumonia group on lung microbiome analysis. Metagenomics analysis of plasma mEVs demonstrated a significant increase in *Cutibacterium* in the non-pneumonia group compared to the pneumonia group.

Interestingly, the correlation between the plasma mEVs and ETA microbiome differed significantly between the pneumonia and non-pneumonia cases (Figure 5). Non-pneumonia subjects exhibited a significantly higher correlation compared to the pneumonia patients. The genus-level analysis of the correlation between the ETA microbiome and plasma mEVs revealed that Acinetobacter and Lactobacillus, with correlation coefficients > 0.5 across sampling sites, were associated with non-pneumonia cases. However, there was no genus associated with pneumonia (Table 2).

## 4. Discussion

Intercellular communication mediated by bacterial EVs is involved in the pathogenesis of lung diseases [2,8]. Previous studies have investigated the migration of EVs derived from gut microbiota to distant sites and their impact on gut permeability [7,11,15]. Emerging evidence suggests the existence of a gut–lung axis, where EVs produced by commensal bacteria in the gut may contribute to mucosal tolerance and protect against lung diseases [16].

However, there is a lack of research on lung microbiome distribution across different body sites and the effects of EVs from other organs on lung inflammation. We found that the lung and plasma microbial communities have distinct compositions and form separate clusters, consistent with previous studies demonstrating weak correlations in the EV composition across different body sites [17,18]. Nah et al. demonstrated that T2DM patients exhibit intestinal permeability dysfunction and higher correlations among stool, serum and urine samples compared to healthy individuals [7]. Based on the increased pulmonary permeability in acute lung injury, such as in ARDS [19], we hypothesized that the correlations between body sites would be lower in healthy subjects compared to pneumonia patients. However, contrary to our expectations, the correlation between the lung microbiome and plasma mEVs was higher in individuals without pneumonia than in pneumonia patients.

We found that the migration of protective bacteria-derived EVs was more likely to increase in the absence of pneumonia, rather than the transfer of EVs derived from respiratory pathogens to the plasma due to increased alveolar-capillary permeability in pneumonia [20].

Macia et al. reported that, under homeostatic conditions, EVs from commensal microbiota suppress pathogenic colonization and regulate the immune response to tolerance [21]. Recent studies have demonstrated that EVs from gram-negative and gram-positive bacteria can have both detrimental and beneficial effects in pneumonia [6].

Outer membrane vesicles (OMVs) interact with the respiratory epithelium and trigger inflammatory responses in the innate and adaptive immune systems during pneumonia [22,23,24,25]. However, OMVs from bacteria, such as *B. pertussis*, *S. pneumoniae*, *K. pneumoniae* and *A. baumannii*, have also been considered as potential vaccine candidates based on their ability to induce robust antibody responses and beneficial biochemical properties [26,27,28,29].

Several studies have demonstrated that native OMVs from *A. baumannii* provide protection against clinical isolates, including pan-resistant strains, and reduce mortality [29,30,31,32]. Cai et al. demonstrated that OMVs from *A. baumannii* can induce dendritic cell activation, promoting Th2 activation and humoral immune responses [33]. Similarly, the increased intercellular communication of Acinetobacter EVs in non-pneumonia cases in our study can be interpreted in terms of the immunogenic and protective potential of EVs.

Our data also revealed that the genus Lactobacillus, with a correlation coefficient > 0.5 between lung and plasma mEVs, was associated with the non-pneumonia status. The beneficial effects of probiotics against infectious diseases are reportedly mediated by probiotic-derived EVs, which inhibit pathogens, enhance barrier function and modulate immunity [34]. Consistent with our findings, previous studies have shown that EVs from different strains of Lactobacillus possess anti-inflammatory properties that are beneficial for hosts [35,36,37].

This study also had some limitations. First, the sample size was relatively small. Second, we were unable to acquire stool samples and could not analyze the microbial community across various body sites. Therefore, we could not confirm whether or not the gut microbiome serves as a major source of plasma microbial EVs. Third, as this study was a cross-sectional clinical study, causality could not be established and the underlying pathophysiological mechanisms could not be fully elucidated. Although some data exist on the beneficial role of mEVs derived from Acinetobacter and Lactobacillus, additional in vivo animal studies are needed to evaluate the migration of microbe-derived EVs across various body sites and their therapeutic effects in pneumonia. Despite these limitations, our study provided a direct comparison between lung microbiome and plasma microbial EVs using human samples, allowing a comprehensive understanding of the microbiome and its communication through EVs.

## 5. Conclusions

In conclusion, we observed a significantly higher correlation between lung microbiome and plasma EVs in non-pneumonia individuals than in pneumonia patients. This indicates the potential of EVs as a tool for host–bacterial communication. Further large-scale translational studies are warranted to further elucidate the role of microbe-derived EVs across various body sites in pneumonia.

## Figures and Tables

**Figure 1 microorganisms-11-02128-f001:**
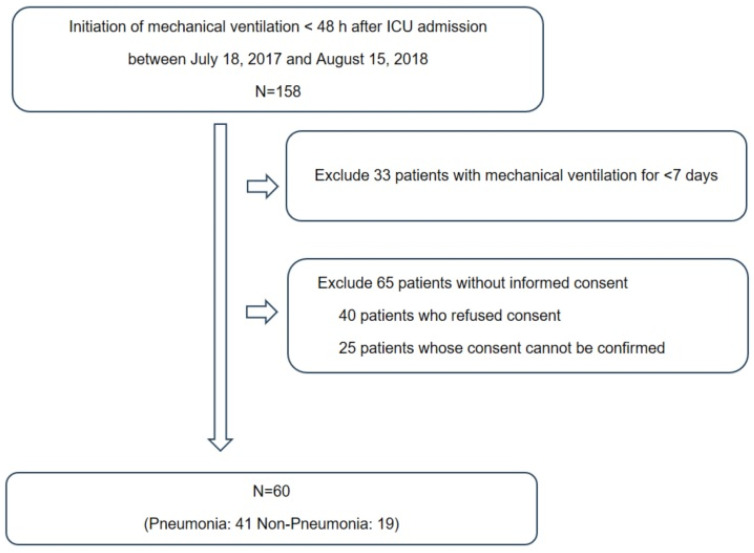
The study participants framework.

**Figure 2 microorganisms-11-02128-f002:**
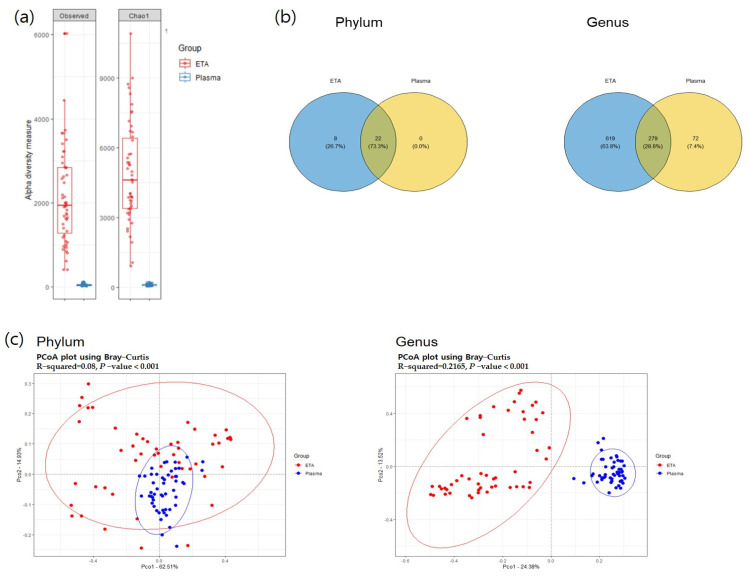
Alpha and beta diversity. (**a**) Observed OTU and Chao1 alpha-diversity analysis of the subjects for comparison of ETA and plasma samples. (**b**) The common phyla and genera in both sites. The overlapping circle in the Venn diagram represents the number of common microorganisms in both sites. (**c**) PCoA analysis visualized on the Bray–Curtis dissimilarity index for beta diversity across body sites.

**Figure 3 microorganisms-11-02128-f003:**
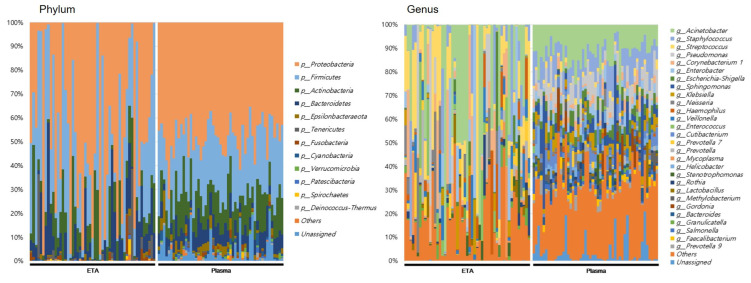
Microbial composition at phylum and genus in intubated patients. The numbers indicate the relative proportion of the microbiome of ETA and plasma mEVs.

**Figure 4 microorganisms-11-02128-f004:**
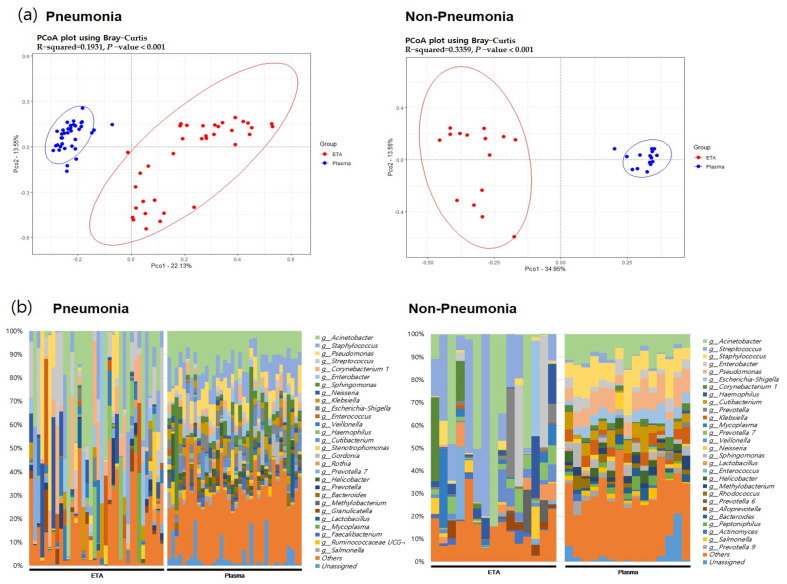
The beta diversity between body sites and microbial composition in pneumonia patients and non=pneumonia individuals. (**a**) PCoA analysis with the Bray–Curtis dissimilarity index between ETA microbiome and plasma mEV in the pneumonia and non-pneumonia groups. (**b**) Microbial composition of ETA and plasma mEVs in the pneumonia and non-pneumonia groups.

**Figure 5 microorganisms-11-02128-f005:**
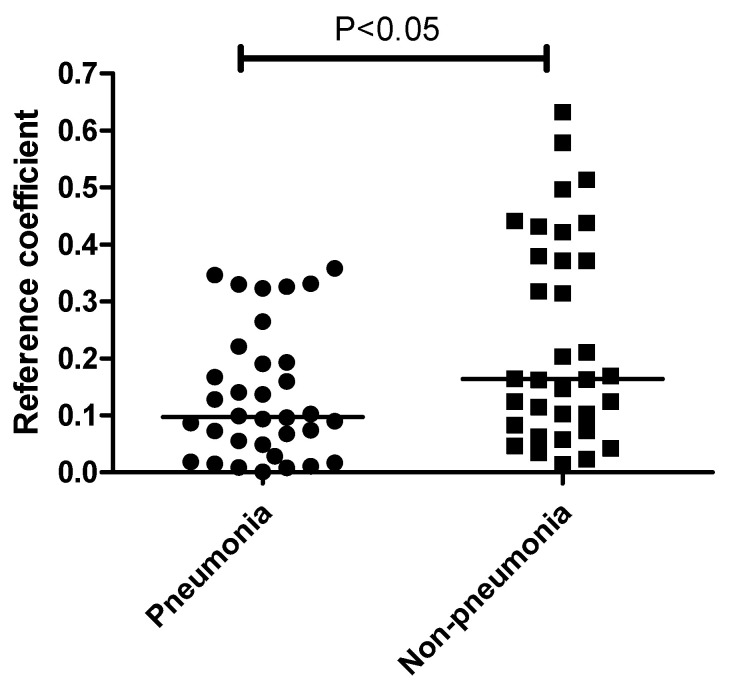
Comparison of correlation coefficients between ETA and plasma mEVs based on the pneumonia status. The correlation coefficients were higher in non-pneumonia group than in the pneumonia group. The line indicates the median level.

**Table 1 microorganisms-11-02128-t001:** Clinical characteristics of the study subjects.

	Pneumonia	Non-pneumonia	*p* Value
	N = 41	N = 19	
Age	73 (59.5, 80.8)	76 (59, 81)	0.732
Male	30 (73.2%)	13 (68.4%)	0.763
ARDS	8 (19.5%)	0 (0%)	0.047
Charlson Comorbidity Index	3 (1, 4)	2 (1, 2)	0.051
Reason for intubation			<0.001
Cardiac arrest	1 (2.4%)	3 (15.8%)	
Neurosurgical condition	5 (12.2%)	14 (73.7%)	
Postoperative status	0 (0%)	1 (5.3%)	
Respiratory distress	35 (85.4%)	1 (5.3%)	
PaO_2_/FiO_2_	235.5 (143.8, 304.8)	430.8 (321.0, 457.5)	<0.001
Severity			
APACHE score	20 (16,24)	22 (17.25)	0.165
SOFA score	7 (5.8, 9.0)	6 (4, 9)	0.227
GCS	8 (6, 11)	6 (5, 9)	0.074
28-day mortality	12 (29.3%)	6 (31.6%)	0.99
In-hospital mortality	19 (46.3%)	6 (31.6%)	0.4
MV duration	12 (7.8, 18.0)	10 (7, 14)	0.238
CRP (mg/L)	122.5 (44.0, 201.3)	61 (4.6, 132.1)	0.012
Procalcitonin (ng/mL)	0.59 (0.28, 3.32)	0.23 (0.04, 0.46)	0.012

Data are expressed as median (interquartile range) unless otherwise indicated. PaO_2_/FiO_2_: ratio of arterial oxygen partial pressure to fractional inspired oxygen; APACHE: Acute Physiology, Age, Chronic Health Evaluation II; SOFA: Sequential Organ Failure Assessment; GCS: Glasgow Coma Scale; MV: mechanical ventilation; CRP: C-reactive protein.

**Table 2 microorganisms-11-02128-t002:** Significant genera between ETA microbiome and plasma mEV.

	Site	Genus	Coefficient	FDR *p*-Value
Total	ETA-serum EV	*Acinetobacter*	0.368	0.008
		*Corynebacterium*	0.326	0.019
		*Neisseria*	0.288	0.041
		*Bifidobacterium*	0.318	0.023
		*Lautropia*	0.276	0.049
Pneumonia	ETA-serum EV	*Acinetobacter*	0.346	0.039
		*Corynebacterium*	0.358	0.032
		*Alloprevotella*	−0.033	0.049
Non-pneumonia	ETA-serum EV	*Acinetobacter*	0.632	0.013
		*Lactobacillus*	−0.578	0.024

ETA: endotracheal aspirates, EV: extracellular vesicle.

## Data Availability

The datasets presented in this study can be found in online repositories. The names of the repository/repositories and accession number(s) can be found at: (https://www.ncbi.nlm.nih.gov/sra/PRJNA796437, and https://www.ncbi.nlm.nih.gov/sra/PRJNA678854, accessed on 1 August 2023).

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
