# Peer review of "Association between the Respiratory Microbiome and Plasma Microbial Extracellular Vesicles in Intubated Patients"

_microorganisms, 2023, doi:10.3390/microorganisms11092128_

Round 1
Reviewer 1 Report
line 59: ETA and plasma samples were collected after > 8 h of fasting. Please provide explanation if fasting is ok for elderly persons and not interfering with the underlying pneumonia disease that's severe enough to be intubated
Abstract: Samples from 51 persons were analyzed (9 persons were excluded because insufficient read depths (line 96). Please specify male vs. females of the excluded participants. Also please indicate in Table 1 whose samples were used for the analysis and whose samples were excluded.
Table1: please explain (mean, SD?)
Line 146 citation for the previous study of 111 patients
Figure 4: I think that the X-axis labels are inverted
Table 2: FDR p-value: there were several p-values below 0.05 in the pneumonia group, please reconcile with impression that there was no correlation in the pneumonia group.
English is fine, the manuscript is carefully written and carefully put together
Author Response
1.line 59: ETA and plasma samples were collected after > 8 h of fasting. Please provide explanation if fasting is ok for elderly persons and not interfering with the underlying pneumonia disease that's severe enough to be intubated
Response 1: We appreciate the reviewer’s comments.
The recent clinical study showed no differences in patient-centered outcomes between the fasting and matched control group. on the very early phase of ICU stay (<72hours).
--Cardozo Júnior LCM, et al.reported that fasting during the first 72 h in ICU (ie, absence of oral nutrition, EN, or PN support) was not associated with worse outcomes compared with early nutrition support.
The result aligns with recent clinical trials that could not demonstrate improvement in patient‐centered outcomes both in the short and longterm when comparing higher vs lower energy goals in the early stages of acute critical illness
<References>
-TARGET Investigators, for the ANZICS Clinical Trials G,Chapman M, Peake SL, Bellomo R, et al. Energy‐dense versusroutine enteral nutrition in the critically ill.NEnglJMed.2018;379(19):1823‐1834.6.
-Arabi YM, Aldawood AS, Haddad SH, et al. Permissive underfeedingor standard enteral feeding in critically ill adults.N Engl J Med.2015;372(25):2398‐2408
There is clinical evidence suggesting that in early phase of critical illness, the provision of nonvolitional feeding is not able to suppress catabolism or reduce muscle loss.
<References>
-Wischmeyer PE. Nutrition therapy in sepsis. Crit Care Clin.2018;34(1):107‐125.24.
-Dresen E, Weißbrich C, Fimmers R, Putensen C, Stehle P. Medicalhigh‐protein nutrition therapy and loss of muscle mass in adult ICUpatients: a randomized controlled trial.Clin Nutr. 2021;40(4):1562‐1570.25.
-Streat SJ, Beddoe AH, Hill GL. Aggressive nutritional support doesnot prevent protein loss despite fat gain in septic intensive carepatients.J Trauma. 1987;27(3):262‐266.26.
-Frankenfield DC, Smith JS, Cooney RN. Accelerated nitrogen lossafter traumatic injury is not attenuated by achievement of energybalance.JPEN J Parenter Enteral Nutr. 1997;21(6):324‐329
- Abstract: Samples from 51 persons were analyzed (9 persons were excluded because insufficient read depths (line 96). Please specify male vs. females of the excluded participants. Also please indicate in Table 1 whose samples were used for the analysis and whose samples were excluded.
Response 2: We appreciate the reviewer’s comments. Corrections have been made to avoid confusion.
“Of 120 samples collected from 60 subjects, nine samples were excluded due to low read counts and 111 samples were finally used in the analysis.”
Each patient included one or two samples, so a total of 60 patients participated, and 111 samples were used.
- Table1: please explain (mean, SD?)
Response 3 We added the footnotes of Table 1.
“Data are expressed as median (interquartile range) unless otherwise indicated. PaO2/FiO2: ratio of arterial oxygen partial pressure to fractional inspired oxygen; APACHE: Acute Physiology, Age, Chronic Health Evaluation II; SOFA: Sequential Organ Failure Assessment; GCS: Glasgow Coma Scale; MV: mechanical ventilation; CRP: C-reactive protein.”
- Line 146 citation for the previous study of 111 patients
Response 4. Corrections have been made to avoid confusion.
“In both pneumonia and nonpneumonia cases, the composition of plasma mEVs was distinct from that of the ETA microbiome, which is consistent with the aforementioned findings in total 111 samples.”
Figure 4: I think that the X-axis labels are inverted
We revised figure 4. To avoid confusion, we revised Y axis from 1-correlation coefficient to correlation coefficient.
- Table 2: FDR p-value: there were several p-values below 0.05 in the pneumonia group, please reconcile with impression that there was no correlation in the pneumonia group.
Response 5: We appreciate the reviewer’s comments.
In the statistical analysis, we defined genera with high correlation between ETA and plasma mEV. Genera with a correlation coefficient exceeding 0.5 and a false discovery rate less than 0.05 were considered statistically significant
In the pneumonia group, some genera with FDR below 0.05 had a correlation below 0.05. Therefore, the genera were considered statistically insignificant

Reviewer 2 Report
The aim of the study is to evaluate the relationship between the pneumonia microbiome and other parts of the body, in this particular case plasma. It is concluded that the correlation found between plasma vesicles in patients without pneumonia represents a beneficial effect of the vesicles and their immune activity.
The introduction is brief, more should be said about extracellular vesicles and their microbiological components.
The study is interesting, but it should be noted that there is an individual response in both healthy patients and patients with pneumonia. Likewise, the immune response is also dependent on the time of disease presentation, the disease process and the consolidation of the disease.
In materials and methods, there is a lot of variability between individuals. Please describe in more detail the type of patient, age, sex and other characteristics necessary for the study, and justify the study design.
In 2.3. a couple of primers are described for the study, but in the results a very large number of genera of microorganisms have been determined. Explain this.
In results:
Below table 1 and 2 explain the acronyms.
Improve the quality of all figures.
I believe that a next step would be to do this same study in vivo experimentally. Trying to avoid all biases.
Author Response
- The introduction is brief, more should be said about extracellular vesicles and their microbiological components.
Response 1: We appreciate the reviewer’s comments. We revised the introduction.
“EVs are known to be enriched with proteins, lipids DNA and miRNAs (4). These EV-containing miRNAs are transferred to specific cells and regulate the pathogenic pro-cess such as angiogenesis, coagulation and inflammation (5). EVs released by respiratory pathogens facilitates proinflammatory responses of pulmonary epithelium and various immune cells (6). In parallel, they can induce robust antibody response and reduce bacte-rial replications as promising vaccine candidates (6).”
2.The study is interesting, but it should be noted that there is an individual response in both healthy patients and patients with pneumonia. Likewise, the immune response is also dependent on the time of disease presentation, the disease process and the consolidation of the disease.
Response 2: We appreciate the reviewer’s comments.
Although there were individual differences within the group, there was a clear difference in the immune response between pneumonia and non-pneumonia. CRP level and procalcitonin level differed between the two groups.
Blood collection and microbiome sample acquisition were performed on the same day within 7 days of starting mechanical ventilation. The CRP level and procalcitonin level were used as representative inflammatory markers. The CRP and procalcitonin had a positive correlation (r=0.449, P=0001). The oxygenation (PaO2/FiO2) and CRP had negative correlation (r=-0.328, P=0.010). The PaO2/FiO2 and Procalcitonin had negative correlation (r= -0.292, P=0.044).
This study is a pilot study on the correlation between the lung microbiome and plasma mEV. In further studies, subgroup analysis according to immune response and prognosis within the pneumonia group will help to elucidate the mechanism.
“Inflammatory markers such as CRP and procalcitonin were significantly higher in pneumonia group than non-pneumonia group.”
- In materials and methods, there is a lot of variability between individuals. Please describe in more detail the type of patient, age, sex and other characteristics necessary for the study, and justify the study design.
Response 3: We appreciate the reviewer’s comments.
Despite the diversity of patient groups, there was no difference in underlying disease, sex, and age between the pneumonia and non-pneumonia groups, and there was a clear difference in inflammation levels. Therefore, we tried to construct a case-control group with minimal confusing factors but distinct immune differences.
“The reasons of mechanical ventilation included the cardiogenic problem (n=4), neurosurgical status (n=19), postoperative care (n=1) and respiratory distress (n=36).”
“There were 43 men (71.7%) and 41 people (68.3%) over the age of 65.”
“The baseline characteristics of the subjects are presented in Table 1. There were 43 men (71.7%) and 41 people (68.3%) over the age of 65. There were 41 pneumonia cases and 19 nonpneumonia cases. Despite the variation in the characteristics of the patients, there were no significant differences between the pneumonia and nonpneumonia groups in terms of age, sex, Charlson Comorbidity Index, severity of illness (SOFA and APACHE II scores), and clinical outcomes, including mortality and duration of mechanical ventilation. Inflammatory markers such as CRP and Procalcitonin were significantly higher in pneumonia group than non-pneumonia group. As expected, the nonpneumonia group exhibited higher PaO2/FiO2 ratios. Regarding the causes of intubation, neurosurgical conditions and cardiac arrest were more frequently identified in the nonpneumonia group”
- In 2.3. a couple of primers are described for the study, but in the results a very large number of genera of microorganisms have been determined. Explain this.
Response 4: We appreciate the reviewer’s comments.
For bacterial identification, 16S rRNA gene sequencing is the most commonly used method. These reasons include (i) its presence in almost all bacteria; (ii) the function of the 16S rRNA gene over time has not changed; and (iii) the 16S rRNA gene (1,500 bp) is large enough for informatics purposes. Using 16S rDNA sequences, numerous bacterial genera and species have been reclassified and renamed,
“In our study, the extracted DNA underwent amplification using primers designed to target the V3 to V4 regions of the prokaryotic 16S rRNA gene. V3–V4 is widely used primer set, targeting different hypervariable regions of the 16S rRNA gene.”
References: J Clin Microbiol. 2007 Sep; 45(9): 2761–2764.
- Below table 1 and 2 explain the acronyms.
Response 5: We appreciate the reviewer’s comments and added the footnotes.
Table 1
Data are expressed as median (interquartile range) unless otherwise indicated. PaO2/FiO2: ratio of arterial oxygen partial pressure to fractional inspired oxygen; APACHE: Acute Physiology, Age, Chronic Health Evaluation II; SOFA: Sequential Organ Failure Assessment; GCS: Glasgow Coma Scale; MV: mechanical ventilation; CRP: C-reactive protein.
Table 2
ETA: endotracheal aspirates, EV: extracellular vesicle
- Improve the quality of all figures.
Response 6: We improved the quality of all figures according to reviewer’s comments.
- I believe that a next step would be to do this same study in vivo experimentally. Trying to avoid all biases.
Response 7: We appreciate the reviewer’s comments. We revised the limitation section.
“Although some data exist on the beneficial role of mEVs derived from Acinetobacter and lactobacillus, additional in-vivo animal studies are needed to evaluate migration of microbe-derived EVs across various body sites and their therapeutic effects in pneumonia.”

Round 2
Reviewer 1 Report
thanks for addressing the concerns
Author Response
We appreciate the careful and thoughtful comments by the reviewers
Reviewer 2 Report
Dear authors:
The authors addressed the questions and suggestions I made earlier, the work has improved considerably. I agree that the article should be accepted.
Author Response

(The authors gave the same response as above.)
